# Optomechanics with a hybrid carbon nanotube resonator

A. Tavernarakis[1], A. Stavrinadis [1], A. Nowak[1], I. Tsioutsios[1], A. Bachtold [1] & P. Verlot [1,2]

In just 20 years of history, the field of optomechanics has achieved impressive progress, stepping into the quantum regime just 5 years ago. Such remarkable advance relies on the technological revolution of nano-optomechanical systems, whose sensitivity towards thermal decoherence is strongly limited due to their ultra-low mass. Here we report a hybrid approach pushing nano-optomechanics to even lower scales. The concept relies on synthesising an efficient optical scatterer at the tip of singly clamped carbon nanotube resonators. We demonstrate high signal-to-noise motion readout and record force sensitivity, two orders of magnitude below the state of the art. Our work opens the perspective to extend quantum experiments and applications at room temperature.

[1] ICFO-Institut de Ciencies Fotoniques, The Barcelona Institute of Science and Technology, 08860 Barcelona, Spain. [2] Université de Lyon, Université Claude Bernard Lyon 1, CNRS, Institut Lumière Matière, 69622 Villeurbanne, France. Correspondence and requests for materials should be addressed to P.V. (email: pierre.verlot@univ-lyon1.fr)

Macroscopic physical systems, that are made of millions of atoms, are weakly sensitive to quantum effects. This is essentially due to the unavoidable interaction with the environment, which can generally be modelled as a thermal reservoir. Such coupling can be viewed as a large number of decoherence channels, proportional to the size of the system, through which the quantum properties are washed out by thermal fluctuations. To overcome this somehow fundamental limitation, experiments investigating macroscopic quantum behaviours have been carried out with small size systems operated at low temperature[1–3]. In particular, optomechanics has recently known a crucial evolution, with the emergence of novel experimental methods enabling to strongly couple coherent light to nano-mechanical resonators[4]. This advance has rushed optomechanics into the quantum regime, with such milestones as ground state laser cooling[5], quantum back-action induced correlations[6], and feedback control close to the Heisenberg limit[7]. However, to date, despite the impressive reduction of the dimensions of opto-mechanical systems[8], all optomechanical experiments operating close to the quantum regime of the optomechanical interaction have required cryogenic temperatures for limiting the effects of thermally induced decoherence.

Recently, an important experimental effort has emerged for further decreasing the effects of thermal decoherence and reaching the quantum regime of the optomechanical interaction at room temperature. The proposed approaches essentially rely on designs enabling to decouple the mechanical device from its environment, by means of phononic engineering[9], thin tethers[10,11], and even by suppressing any physical connection to the thermal bath using ultra-high vacuum optical trapping[12,13].

Here we report a hybrid approach enabling unprecedentedly low thermal decoherence level for a solid-state mechanical reso-nator at room temperature. The concept is to selectively grow an efficient optical scatterer at the tip of singly clamped, micrometre-long carbon nanotube (CNT) resonators. Because of their ultra-low mass, we show that our devices are 200 times less sensitive to thermal noise than the recently reported state of the art[10,11,14], reaching levels that were previously confined to cryogenic environments[15,16]. Our work appears as an excellent way to use nanoparticles as scanning sensors, opening the perspective of enhanced performances in various fields such as surface ima-ging[17,18], magnetic[19,20] and force[21,22] microscopy as well as for unprecedentedly sensitive cavity optomechanical studies[23].

## Results

**Hybrid CNT nano-optomechanical device.** The hybrid device we fabricate and use in the present work is shown in Fig. 1. It consists of singly clamped suspended carbon nanotube resonators at the tip of which a nanoparticle is subsequently grown. The carbon nanotubes are grown via chemical vapour deposition from the edges of n-doped silicon wafers[24], yielding to nanotubes with diameters typically in the 1−3 nm range. Fabrication of the Pt nanoparticle on the suspended nanotube tip and sample imaging were achieved using a Zeiss FIB (Zeiss Auriga 60 FIB-SEM), 1-nm resolution GEMINI scanning electron microscope (SEM) equip-ped with a gas injection system (GIS). The baseline pressure of the sample chamber typically ranges from $8 \times 10^{-7}$ to $2 \times 10^{-6}$ mBar. For fabricating the Pt nanoparticle, the nozzle of the GIS is positioned at approximately $\leq 100\,\mu$m from the sample and the gas precursor used is Trimethyl(methylcyclopentadienyl)plati-num(IV). Upon injection of the precursor gas in the sample

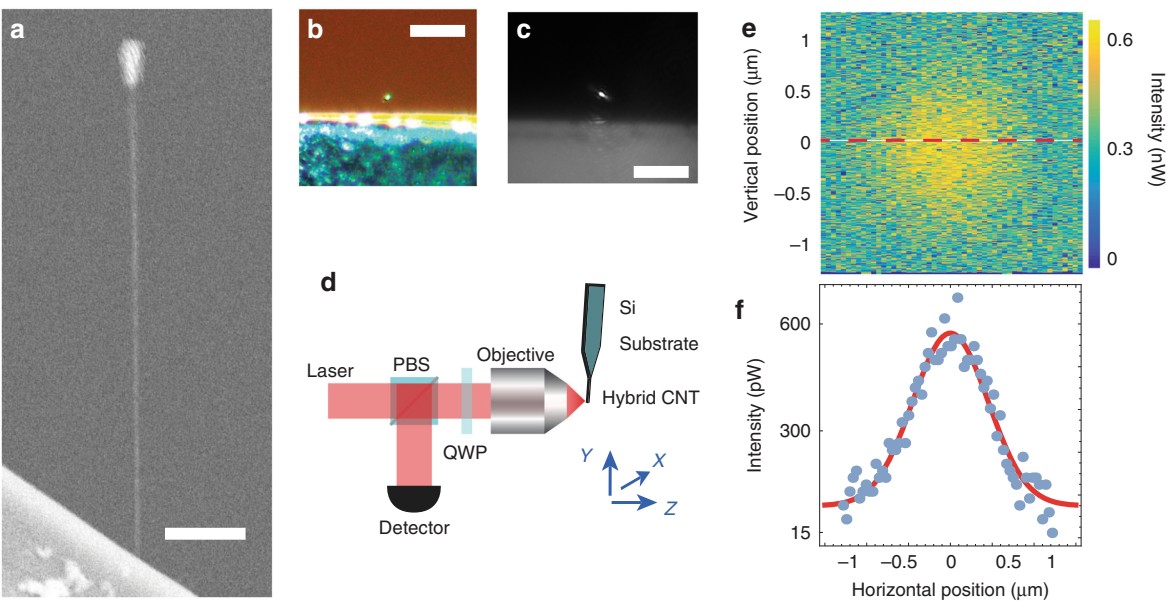

**Fig. 1** Hybrid carbon nanotube nano-optomechanical device and experimental setup. **a** Scanning electron micrograph showing the hybrid carbon nanotube resonator used in the present work. The n-doped silicon wafer on which the device is clamped appears at the bottom. The scale bar is 1 μm. **b** Optical micrograph showing the device used in this work. Picture obtained using white light illumination and a ×50 magnification. The Pt nanoparticle appears as a very bright spot (top of the image). The scale bar is 20 μm. **c** Optical micrograph of the sample as directly mounted on our experiment. 632 nm laser light and an aspherical lens with numerical aperture NA = 0.55 are used. The scale bar is 20 μm. **d** Schematic of the experimental setup. The sample is placed at the waist ($Z = 0$) of a strongly focused beam of coherent light delivered by a He-Ne laser. The scattered light is collected in reflection by means of an optical circulator and further sent on an avalanche photodetector. Both the sample and focusing system are mounted in a vacuum chamber. **e** Scattered intensity collected in reflection as a function of the position of the Pt particle in the beam waist. A 2D Gaussian distribution is observed. **f** Reflected intensity at the centre of the beam waist (across the dashed red line from Fig. 2e) as a function of the transverse position. A Gaussian fit yields to a waist $w_{bs} = 900$ nm, matching the incident beam waist $w_0$

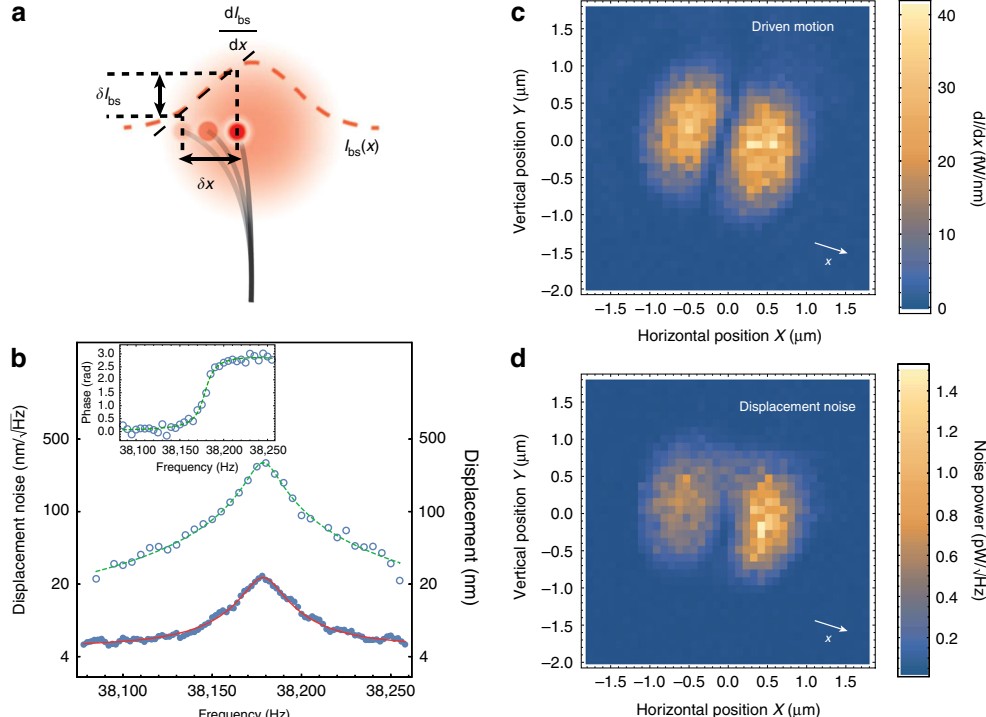

**Fig. 2** Nano-optomechanical coupling. **a** Schematic illustrating the nano-optomechanical coupling to a focused laser probe. The nanoparticle (red dots) scatters light, proportional to the local intensity of the input laser beam (red, dashed line). Consequently, the nanomechanical motion $\delta x$ of the carbon nanotube resonator (black lines) results in fluctuations $\delta I_{bs}$ of the scattered intensity $I_{bs}$, proportional to the intensity gradient $dI_{bs}/dx$. **b** Driven response and noise spectrum of the hybrid CNT device. Circles represent the calibrated displacements induced by a frequency swept piezo drive. The inset represent the corresponding phase response. The dashed lines are theoretical fits assuming single harmonic oscillator. Blue dots represent the calibrated motion spectral density in absence of any motion driving source. Solid line corresponds to a Lorentzian fit, as expected for a thermally driven, weakly damped harmonic oscillator. **c** Modulation of the scattered intensity as a function of transverse and vertical position in the beam waist. A constant piezo drive is applied at the mechanical resonance frequency of the hybrid device. The beam waist is subsequently scanned in the transverse plane (X, Y) using a step motor stage, and the corresponding intensity modulation is recorded using a low noise lock-in amplifier. The obtained intensity distribution identifies to the intensity gradient in the motion direction x (white arrow), as depicted in **a**. **d** Fluctuations of the scattered intensity as a function of transverse and vertical position in the beam waist. A similar intensity distribution to that of the piezo-driven case is obtained, due to Brownian motion of the hybrid nano-optomechanical device

chamber, the pressure is increased to values typically ranging from $8 \times 10^{-6}$ to $2.5 \times 10^{-5}$ mBar. In the meantime, the region of the nanotube on which Pt is deposited is scanned by the electron beam. The time required for the Pt deposition to be completed depends on the intended size of the Pt nanoparticle and typically ranges from 3 to 10 s. Note that our method can be extended to a variety of materials, including dielectrics for purely dispersive optomechanical applications[4].

Our experimental setup is schematically depicted in Fig. 1d. A single mode helium-neon laser delivering coherent light at a wavelength $\lambda = 632$ nm is focused on the sample by means of an aspherical lens with numerical aperture NA = 0.55. The back-scattered intensity $I_{bs}$ is collected by means of an optical circulator followed by a high gain avalanche photodetector (see Fig. 1d). Scanning the sample across the focal plane enables to reconstruct the back-scattered intensity profile (Fig. 1e,f). A 2-dimensional Gaussian distribution is revealed, $I_{bs}(r) = \eta_c \frac{2\sigma_{scatt}}{\pi w_0^2} I_0 e^{-\frac{2r^2}{w_{bs}^2}}$ ($\eta_c$ the collection efficiency, $\sigma_{scatt}$ the scattering cross section, $I_0$ the incident photon flux, $r$ the distance to the axis of the beam) from which an effective width $w_{bs} = 900$ nm can be extracted, matching the waist $w_0$ of the incident beam. This is consistent with the small size parameter approximation $d/2 \ll \lambda$ ($d$ the characteristic diameter of the particle), the optical scatterer acting as a local probe of the electromagnetic field. From the

intensity scattered at the centre of the beam $I_{bs}(r=0) = 1.2 \times 10^{-3} I_0$, it is possible to estimate the particle size: Assuming spherical scattering geometry, the collection efficiency is given by $\eta_c = \frac{1}{\pi} \arcsin \frac{NA}{n_a} \simeq 0.12$ ($n_a \simeq 1.5$ the refractive index of the asphere), yielding to $\sigma_{scatt} \simeq 1.28 \times 10^{-14}$ m².

**Optomechanical coupling**. Combined to the very large motion of carbon nanotube resonators, the above outlined characteristics make our scheme intrinsically sensitive to the displacement fluctuations of our nanomechanical device. The principle is depicted on Fig. 2a: When shifted from the centre of the beam, the motion fluctuations of the resonator result in a modulation of the back-scattered intensity, proportional to the displacements. The strongly focused nature of the incident beam results in large intensity gradients, enabling very high motion sensitivity. Figure 2b shows both piezo-driven (circles) and free running (dots) nano-optomechanical variations in the Fourier space. The driven curve shows a resonant behaviour, compatible with a harmonic oscillator (dashed line) with mechanical resonance frequency $\Omega_m / 2\pi = 38,178.5$ Hz and quality factor $Q_m = 2245$. The inset shows the corresponding phase response. Moreover, even with no external drive being applied, the spectrum of the nano-optomechanical signal reveals the presence of a resonant peak (dots in Fig. 2b) with the same width and resonance frequency as the one inferred from the piezo-driven response. Further

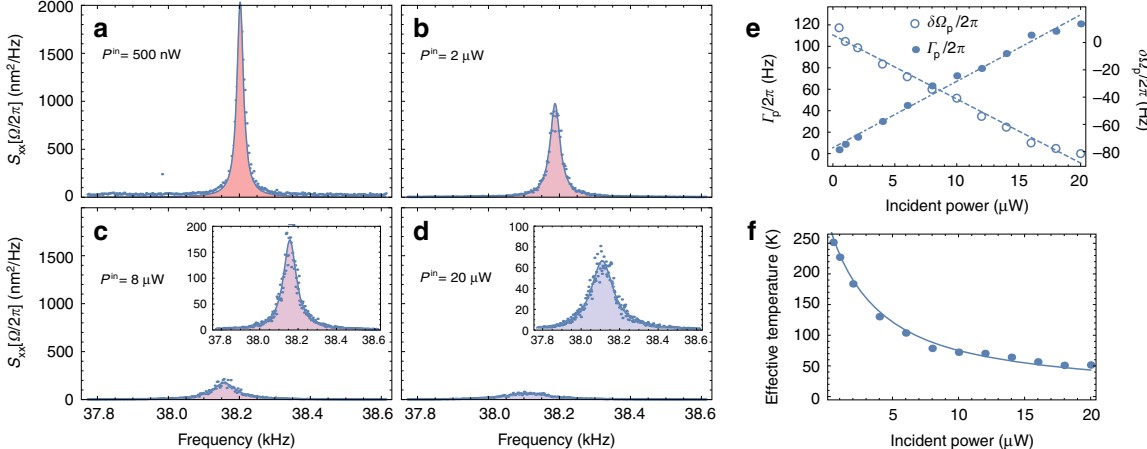

**Fig. 3** Dynamical back-action and photothermal cooling. **a–d** Evolution of the motion spectral density as a function of the input power. The Hybrid CNT is positioned in the middle of the left detection lobe ($x = -w_0/2$, $y = 0$, $z = 0$), where the intensity gradient is maximum. Panels **a–d** are acquired for an input power of 0.5, 2, 8 and 20 μW, respectively (blue dots are experimental data). Insets in **c**, **d** show a magnified view of the curves shown on **c** and **d**, respectively. The straight lines correspond to Lorentzian fits from which the effective mechanical resonance frequencies and damping rates are inferred. A strong decrease of the integrated spectral density (motion variance, colour shaded areas) is observed, corresponding to cooling of mechanical motion. **e** Evolution of the damping rate $\Gamma_p$ (blue dots) and frequency shift $\delta\Omega_p$ (circles) associated with the photothermal gradient. Linear evolutions are obtained, compliant with a linear dynamical back-action model (dashed and dashed-dotted lines). **f** Effective motional temperature as inferred from integrating the motion spectra (blue dots). The solid line corresponds to a measurement back-action-free model, from which we conclude that laser-induced fluctuations are negligible

calibrating these nanomechanical displacements yields to the variance of the associated motion fluctuations $\Delta x^2 = (293\,\text{nm})^2$ (see Methods). Assuming that this displacement noise is of a thermal origin (which will be further confirmed), the equipartition theorem can be used in order to estimate the effective mass of the resonator $m_{\text{eff}} = \frac{k_B T}{\Delta x^2 \Omega_m^2} \simeq 0.79 \times 10^{-18}$ kg ($k_B$ Boltzmann's constant and $T$ the temperature of the environment). As explained in Supplementary Note 2, this value essentially reflects that of the Pt nanoparticle's mass which we estimate to be on the order of $m_{\text{Pt}} \simeq 0.62 \times 10^{-18}$ kg.

As depicted on Fig. 2a, the motion sensitivity is determined by the local intensity gradient. This is verified by measuring the nano-optomechanical response to an off-resonant, piezo-driven actuation: The piezo drive creates a modulation of the position $\delta x_{\text{drive}}(t)$, yielding to a modulation of the back-scattered intensity $\delta I_{\text{bs}}(t) \simeq \frac{\partial I_{\text{bs}}}{\partial x} \delta x_{\text{drive}}(t)$ which is further sent into a lock-in amplifier. Figure 2c shows the result obtained experimentally as we displace the tip of the nanotube in the focal plane ($X$, $Y$) ($X$ (resp. $Y$) the horizontal (resp. vertical) coordinate of the nanopositioning motor stage). As expected, the nano-optomechanical transduction is proportional to the derivative of the intensity profile along the displacement direction: The sensitivity is minimal along the symmetry axis of the optical mode, whereas it is maximal at the centre of each two symmetric lobes. We also repeated the experiment with no external actuation being applied, the nanomechanical motion being driven only by the thermal force. The corresponding intensity fluctuations are recorded by a spectrum analyser. The result is shown on Fig. 2d, very similar to Fig. 2c. However, a closer inspection reveals a slight asymmetry between the two detection lobes. Despite the very low input optical power used ($\simeq 1\,\mu$W), this suggests the presence of dynamical effects from the optical beam, leading to either cooling or amplification of the thermal vibrations in the left and right part of the focal plane.

**Photothermal back-action cooling**. The above results have been obtained while maintaining the input optical power as low as possible in order to minimise the potential impact of

optomechanical effects. In the following, we gradually increase the incident optical power and analyse the resulting dynamical consequences. The tip of the nanomechanical resonator is placed at the centre of the left detection lobe, where the optomechanical gradient is maximum ($x = -w_0/2$, $y = 0$, $z = 0$, see Fig. 1 for notations) and the nano-optomechanical fluctuations are acquired using a spectrum analyser while increasing the laser power. The resulting calibrated spectra are shown on Fig. 3a–d for four distinct values of the incident power. A clear reduction of the effective temperature, proportional to the spectrum area, is observed.

Plotting the effective mechanical resonance frequency and damping rate as a function of laser power (Fig. 3e, circles and dots, respectively) reveals linear trends, characteristic of linear dynamical back action[25], which can be explained by the presence of strong delayed force gradients[26] as recently shown in the context of nano-electromechanics[27]. The large displacements of the nanotube resonator in the focal plane result in a motion-dependant optical force (also called ponderomotive force), leading to an effective mechanical response $\chi_{\text{eff}}$ given by:

$$\chi_{\text{eff}}^{-1}[\Omega] = \chi^{-1}[\Omega] - \sum_j H_j[\Omega] \frac{\partial F_j}{\partial x}\bigg|_{x_0}, \qquad (1)$$

with $x_0$ the equilibrium position, $\chi[\Omega] = 1/m_{\text{eff}}(\Omega_m^2 - \Omega^2 - i\Gamma_m\Omega)$ the intrinsic mechanical susceptibility ($\Gamma_m = \Omega_m/Q_m$ the intrinsic mechanical damping rate and $\Omega/2\pi$ the Fourier frequency), $F_j$ the force associated with the $j$th optical process and $H_j[\Omega]$ the Fourier transform of the response function taking into account any delay in the application of $F_j$[28].

In our system, two optical processes must be retained a priori, that are the gradient force $F_{\text{grad}}$ and the photothermal force $F_p$[26,28]. Note that we have neglected the scattering force (that is the force resulting from the momentum exchange between light and mechanical motion), whose radial component vanishes at the beam waist. The gradient force is quasi-instantaneous ($H_{\text{grad}} \simeq 1$) and arises from the strongly focused nature of the input laser beam[29], proportional to the real part of the complex

polarizability of the Pt particle $\alpha$[30], $F_{\mathrm{grad}} = -\frac{2\Re\{\alpha\}x_0 e^{-\frac{2x_0^2}{w_0^2}}}{(\pi w_0)^4} \times \hbar k I_0$ (with $k = 2\pi/\lambda$ the wave vector). Assuming comparable absorption and scattering cross sections ($\sigma_{\mathrm{scatt}} \simeq \sigma_{\mathrm{abs}}$)[31], the real part of the polarizability can be approximated as $\Re\{\alpha\} \approx 2\sigma_{\mathrm{scatt}}/k$[32,33]. On the other hand, the photothermal force results from the partial conversion of electromagnetic energy into heat, leading to structural deformations that are equivalent to nanomechanical displacements. The corresponding motion transduction is typically delayed by the heat diffusion time $\tau$, which is generally modelled by a first order function $H_{\mathrm{p}}[\Omega] = 1/(1 - i\Omega\tau)$.

In the limit of a high mechanical quality factor, the effective mechanical susceptibility $\chi_{\mathrm{eff}}$ remains of a Lorentzian nature, with effective damping rate $\Gamma_{\mathrm{eff}}$ and mechanical resonance frequency $\Omega_{\mathrm{eff}}$ given by:

$$\Gamma_{\mathrm{eff}} = \Gamma_{\mathrm{m}} + \Gamma_{\mathrm{p}}, \tag{2}$$

$$\Omega_{\mathrm{eff}} = \Omega_{\mathrm{m}} + \delta\Omega_{\mathrm{p}} + \delta\Omega_{\mathrm{grad}}, \tag{3}$$

$$\Gamma_{\mathrm{p}} = \frac{1}{m_{\mathrm{eff}}\Omega_{\mathrm{m}}} \Im\{H_{\mathrm{p}}[\Omega]\} \left.\frac{\partial F_{\mathrm{p}}}{\partial x}\right|_{x_0}, \tag{4}$$

$$\delta\Omega_{\mathrm{p}} = -\frac{1}{2m_{\mathrm{eff}}\Omega_{\mathrm{m}}} \Re\{H_{\mathrm{p}}[\Omega]\} \left.\frac{\partial F_{\mathrm{p}}}{\partial x}\right|_{x_0}, \tag{5}$$

$$\delta\Omega_{\mathrm{grad}} = -\frac{1}{2m_{\mathrm{eff}}\Omega_{\mathrm{m}}} \left.\frac{\partial F_{\mathrm{grad}}}{\partial x}\right|_{x_0}. \tag{6}$$

Since the average position of the resonator is set to $x_0 = -w_0/2$, the contribution of the gradient force to the effective mechanical resonance frequency vanishes, $\delta\Omega_{\mathrm{grad}} \propto (\partial F_{\mathrm{grad}}/\partial x)_{x_0} = (1 - 4x_0^2/w_0^2)F_{\mathrm{grad}}(x_0)/x_0 = 0$. Using the above given expression of $H_{\mathrm{p}}$ Eqs. (4) and (5) simplify to $\Gamma_{\mathrm{p}} \simeq \Omega_{\mathrm{m}} \frac{\Omega_{\mathrm{m}}\tau}{1+\Omega_{\mathrm{m}}^2\tau^2} \frac{k_{\mathrm{p}}}{k_{\mathrm{m}}}$ and $\delta\Omega_{\mathrm{p}} \simeq -\Omega_{\mathrm{m}} \frac{k_{\mathrm{p}}/(1+\Omega_{\mathrm{m}}^2\tau^2)}{2k_{\mathrm{m}}}$, where we have noted $k_{\mathrm{p}} = (\partial F_{\mathrm{p}}/\partial x)_{x_0}$ and $k_{\mathrm{m}} = m_{\mathrm{eff}}\Omega_{\mathrm{m}}^2$. Figure 3e shows the linear evolutions of $\Gamma_{\mathrm{p}}$ and $\Omega_{\mathrm{p}}$ as functions of the input optical power, which are correctly described by the previous expressions (provided that the photothermal force is proportional to the input optical power, $F_{\mathrm{p}} \propto \sigma_{\mathrm{abs}}I_0$), with $k_{\mathrm{p}} > 0$. Note that moving the average position of the resonator to the other side of the waist ($x_0 = w_0/2$) changes the sign of $k_{\mathrm{p}}$ according to Eqs. (4) and (5), moving the average position of the resonator to the other side of the waist ($x_0 = w_0/2$) leads to the exact reversed $k_{\mathrm{p}} < 0$, which we have also verified (Supplementary Note 3 and Supplementary Fig. 1).

Moreover, the ratio of the slopes gives direct access to the delay time $\tau$, $\Gamma_{\mathrm{p}}/\delta\Omega_{\mathrm{p}} = -2\Omega_{\mathrm{m}}\tau$, from which we infer $\tau = 2.75\,\mu s$. Such value rather indicates a dynamical back action mechanism of a thermal nature, as opposed to optical processes which are much faster. The effective thermal conductivity associated with thermal transport in the longitudinal direction can be estimated $\kappa_{\mathrm{eff}} \simeq \rho_{\mathrm{c}}L^2 c_{\mathrm{p}}/\tau \simeq 12.7\,\mathrm{W\,m^{-1}\,K^{-1}}$ (with $L \simeq 5\,\mu m$ the length of the carbon nanotube resonator and $\rho_{\mathrm{c}} \simeq 2000\,\mathrm{kg\,m^{-3}}$ and $c_{\mathrm{p}} \simeq 700\,\mathrm{J\,K^{-1}\,kg^{-1}}$ the mass density and specific heat of graphite, respectively). This value is much reduced compared to previous reports which have pointed the extraordinarily large thermal conductivity of carbon nanotubes at room temperature[34], on the order of $\kappa_0 \simeq 1000\,\mathrm{W\,m^{-1}\,K^{-1}}$[35]. One possible explanation is that the observed thermal behaviour results from the presence of an amorphous carbon shell, which acts as a very efficient thermal

insulator. From the ratio $\kappa_{\mathrm{eff}}/\kappa_0$, a shell thickness on the order of 3.9 nm can be determined, consistent with independent, e-beam assisted dynamical measurements[36] (Supplementary Note 1). Another possible contribution to the low measured thermal conductivity might be related to enhanced phonon scattering along the nanotube. Indeed, disorder or contamination along the nanotube might be produced during the electron beam-induced growth of the nanoparticle.

Importantly, the hereby reported photothermal cooling mechanism is general, characteristic of the presence of dissipation (both optical and mechanical) in our system. This effect can be enhanced for obtaining native, strong thermal noise cancellation that may be of primary interest, e.g. in cavityless nanomechanical sensing experiments[21,22]. Some of the key parameters involved in the efficiency of this cooling scheme include the absorbance of the nanoparticle, the quality of the thermal contact with the nanotube resonator, and anisotropic thermal expansion (which is the main conversion mechanism of heat into mechanical motion), which can all be further engineered. Concurrently, enhancing dissipation leads to increased back action and readout noises, resulting in an increased measurement uncertainty compared to the purely dispersive case, which is detrimental for quantum physical purposes and notably limits coherent cooling performances well above the quantum regime[7,37,38]. To this end, one may rather minimise dissipation (e.g. by selecting a different optical material and further optimising the nanofabrication and nano-engineering processes) and design alternative schemes adapted to quantum-limited coherent motion control[4,39].

**Force sensitivity and quantum limits**. As already mentioned above, the effective temperature is deduced from the area of the displacement spectrum which identifies the motion variance $\Delta x_{\mathrm{eff}}^2 = \int_{-\infty}^{+\infty} \frac{d\Omega}{2\pi} S_{\mathrm{xx}}^{\mathrm{eff}}[\Omega] \simeq \frac{S_{\mathrm{FF}}^{\mathrm{eff}}}{2m_{\mathrm{eff}}^2\Omega_{\mathrm{eff}}^2\Gamma_{\mathrm{eff}}}$, with $S_{\mathrm{xx}}^{\mathrm{eff}}[\Omega]$ the displacement spectrum and $S_{\mathrm{FF}}^{\mathrm{eff}}$ the broadband spectral density of the total force driving the nanotube resonator. In general, this force is the sum of two uncorrelated contributions, the thermal noise with spectral density given by the fluctuation-dissipation theorem $S_{\mathrm{FF}}^{\mathrm{th}} = 2m_{\mathrm{eff}}\Gamma_{\mathrm{m}}k_{\mathrm{B}}T$ and the back-action noise with spectral density $S_{\mathrm{FF}}^{\mathrm{ba}} = 2m_{\mathrm{eff}}\Gamma_{\mathrm{m}}k_{\mathrm{B}}T_{\mathrm{ba}}$ which depends on the probe power, $T_{\mathrm{ba}} = R_{\mathrm{ba}}P$, with $R_{\mathrm{ba}}$ a coefficient that can be viewed as an effective quantum back-action resistance (in $\mathrm{K\,W^{-1}}$) and $P = \frac{2\pi\hbar c}{\lambda}I_0$ the incident optical power. The temperature is inferred from the equipartition theorem, $T_{\mathrm{eff}} = k_{\mathrm{m}}\Delta x_{\mathrm{eff}}^2/k_{\mathrm{B}}$, yielding to $T_{\mathrm{eff}} = \frac{\Gamma_{\mathrm{m}}}{\Gamma_{\mathrm{eff}}}(T + T_{\mathrm{ba}})$. The effective temperature is plotted on Fig. 3f (dots), together with the latter theoretical expression (solid line), where $R_{\mathrm{ba}}$ is set to $0\,\mathrm{K\,W^{-1}}$, with no additional free parameter being used. The very good agreement of the experimental data to our model demonstrates that the contribution of measurement induced heating is negligible, $T_{\mathrm{ba}} \ll T$, which defines a cold damping mechanism[40]. This result directly establishes the thermal noise as the main source of motion imprecision over the nanomechanical motion detection bandwidth, with a corresponding force sensitivity given by the equivalent input noise of the force estimator, $S_{\mathrm{FF}}^{\min}[\Omega] = S_{\mathrm{xx}}^{\mathrm{eff}}[\Omega]/|\chi_{\mathrm{eff}}[\Omega]|^2 \simeq S_{\mathrm{FF}}^{\mathrm{th}}[\Omega] = 2m_{\mathrm{eff}}\Gamma_{\mathrm{m}}k_{\mathrm{B}}T \approx (767\,\mathrm{zN}/\sqrt{\mathrm{Hz}})^2$, more than two orders of magnitude below the state of the art at room temperature[10,11,14].

Interestingly, our results enable to accurately estimate the photothermal back-action force noise induced by the optomechanical measurement. The photothermal force is proportional to the local intensity, $F_{\mathrm{p}}(x) = F_{\mathrm{p}}(x = 0)e^{\frac{-2x^2}{w_{\mathrm{bs}}^2}}$, with $F_{\mathrm{p}}(x = 0)$ the force at the centre of the beam waist. The partial derivative of the force is therefore given by $(\partial F_{\mathrm{p}}/\partial x)_{x_0} = \frac{-4x_0}{w_0^2}F_{\mathrm{p}}(x_0)$: Because of

the Gaussian geometry, the force gradient gives a direct access to the optical force, $F_p(x_0) = -\frac{w_0^2}{4x_0} \times (\partial F_p/\partial x)_{x_0}$. Combining the latter expression with Eq. (5), we determine $F_p(x_0) \simeq \frac{w_0}{2} \times (1 + \Omega_m^2 \tau^2) \times 2m\Omega_m \delta\Omega_p = 1.2 \times 10^{-16}\,\mathrm{N}$ in the middle of the left detection lobe ($x_0 \simeq -w_0/2$) and for an incident optical power $P \simeq 20\,\mu\mathrm{W}$. The corresponding quantum back-action noise, arising from the granularity of light, is obtained as $S_{FF}^{p,ba} = (F_p/\sqrt{I_0})^2 = (1.5 \times 10^{-23}\,\mathrm{N}/\sqrt{\mathrm{Hz}})^2$. It is interesting to compare the photothermal back-action force to the optical gradient force, which represents the fundamental back-action mechanism (i.e in absence of any absorption) associated with the optomechanical measurement at the beam waist. Using the expression given above, we obtain $F_{grad} \simeq 5.8 \times 10^{-17}\,\mathrm{N}$ and $S_{FF}^{grad,ba} = (F_{grad}/\sqrt{I_0})^2 = (7.1 \times 10^{-24}\,\mathrm{N}/\sqrt{\mathrm{Hz}})^2$ for $P \simeq 20\,\mu\mathrm{W}$ and $x_0 \simeq -w_0/2$: Despite the quite large absorption $\simeq 50\%$, the photothermal drive has a limited effect, barely twice as large compared to the fundamental gradient force. This is quite unusual in optomechanical systems, and may be explained by the very-high level of symmetry of our device, which strongly limits polymorphous-induced inhomogeneous thermal expansion[41].

Finally, we characterise the quantum efficiency of the optomechanical measurement by evaluating the uncertainty product $\sqrt{S_{xx}^{imp} S_{FF}^{ba}}$, with $S_{xx}^{imp}$ denoting the equivalent motion imprecision. From the measurement background level, we obtain $S_{xx}^{imp} \simeq (129\,\mathrm{pm}/\sqrt{\mathrm{Hz}})^2$, limited by the electronic noise of the avalanche photodetector. We subsequently find $\sqrt{S_{xx}^{imp}(S_{FF}^{p,ba} + S_{FF}^{grad,ba})} \simeq 57 \times \frac{\hbar}{2}$. While this value may appear quite large, it represents a 3000-fold improvement over electron beam electromechanical detection[27] which is so far the only existing alternative for carbon nanotube nanomechanical noise detection at room temperature[36]. Moreover, the origin of such imprecision can be accurately addressed in our experiment: As mentioned above, the back-action excess resulting from photon absorption is on the order of $(S_{FF}^{p,ba} + S_{FF}^{grad,ba})/S_{FF}^{grad,ba} \simeq 3$. The rest of the uncertainty product essentially arises from detection imperfections, which increase the measurement imprecision by a factor $1/\sqrt{\eta_{abs}\eta_c\eta_{ph}} \simeq 13$, with $\eta_{abs} \simeq \frac{\sigma_{abs}}{\sigma_{abs}+\sigma_{scatt}} \simeq 0.5$ the probability for an interacting photon to be absorbed, $\eta_c \simeq 0.12$ the collection efficiency and $\eta_{ph} = 0.1$ the Johnson–Nyquist limited photodetector quantum efficiency. By selecting a material with negligible absorption ($\eta_{abs} \simeq 0$, $S_{FF}^{p,ba} \ll S_{FF}^{grad,ba}$), changing the aspherical input lens by a high numerical aperture objective ($\eta_c \simeq 1/\pi$) and selecting a better photodetector, an uncertainty product as low as $\sqrt{S_{xx}^{imp} S_{FF}^{grad,ba}} \simeq 1.8 \times \frac{\hbar}{2}$ should be in reach.

## Discussion

The ability to detect our hybrid carbon nanotube resonator close to the Heisenberg limit is an important step towards quantum optomechanical operation at room temperature, including quantum non-demolition measurements[42] and optomechanical squeezing of coherent light fields[43]. It also represents an indispensable prerequisite for coherent optomechanical preparation and manipulation of quantum nanomechanical states[7,8]: As already mentioned, operating above the Heisenberg limit results in added classical noise within the optomechanical measurement process, which notably yields to a minimal phonon occupation proportional to that noise[44]. Note that the latter perspective will require further improvements of our device, whose Qf product remains limited[8], notably because of the low mechanical

frequency which was selected for high force sensitivity. Enhancing the potential of our approach to quantum mechanical operation may be achieved by selecting resonators with much higher mechanical resonance frequencies[15,36] and further implementing them into a cavity optomechanical design for increasing their Qf product, e.g., by means of strong optical restoring forces[45,46], as was recently proposed in the context of tethered micro-resonators[47].

In conclusion, we have reported a hybrid nano-optomechanical device consisting of a carbon nanotube resonator with a Pt nanoparticle attached at its tip. Such system enables ultra-sensitive detection of the thermally driven vibrations of a carbon nanotube resonator at room temperature, with the additional benefit of ultra-low laser probe power, below 1 μW. Due to the absorptive nature of platinum, we have demonstrated cavityless photothermal cooling of a nanomechanical resonator, which can be further used for significant force sensing enhancement[39]. We have shown that such cooling comes with no significant back-action force, which has enabled us to identify thermal noise as the dominant source of motion imprecision and to subsequently evaluate the force sensitivity on the order of ($767\,\mathrm{zN}/\sqrt{\mathrm{Hz}}$), that is more than two orders of magnitude below the current state of the art. On the basis of the force measurement and motion calibration of our system, we have been able to determine the Heisenberg uncertainty product of our system on the order of $57\frac{\hbar}{2}$, that is a factor of 3000 below the state of the art for carbon nanotube resonators. Finally, we have shown that the corresponding quantum measurement efficiency is essentially limited by optical losses mechanisms, whose suppression opens the realistic perspective of Heisenberg-limited position detection of carbon nanotube-based nanomechanical resonators.

Importantly, because of its scanning probe geometry, our system has the potential to be employed for a number of applications besides optomechanics, including mass spectroscopy[48], ultra-sensitive force measurements[39,49], scanning probe microscopy[21,22] and ultra-sensitive magnetic imaging[19,20]. Our concept, the deposition of a nanoparticle at the tip of a carbon nanotube, also presents a strong potential for the field of nanoplasmonics[50].

## Methods

**Sample fabrication.** The nanotubes used in this work are grown via chemical vapour deposition on silicon substrates. Nanotubes are attached to the surface of the substrate by van der Waals forces. Some of the nanotubes extend beyond the substrate edges, thus forming singly clamped resonators (Fig. 1b, c), with lengths in the 100 nm–10 μm range.

**Motion calibration.** The calibration of the displacements of the hybrid CNT device essentially relies on comparing the motion variance to the spot size area $w_0^2$. We first position the nanoresonator tip exactly at the centre of the beam waist ($x = 0$, $y = 0$, $z = 0$) and apply a resonant piezo drive, yielding to a strong modulation of mechanical motion $x(t) = x_{cal} \cos(\Omega_m t)$, with $\Omega_m$ the mechanical resonance frequency of the hybrid CNT resonator and $x_{cal}$ the amplitude of the piezo excitation. This motion results in a modulation of the scattered intensity,

$I_{bs}^{on}(t) = \eta_c \frac{2\sigma_{scatt}}{\pi w_0^2} I_0 \exp^{-\frac{2x^2(t)}{w_0^2}}$, which can be expanded to second order as

$I_{bs}^{on}(t) = \eta_c \frac{2\sigma_{scatt}}{\pi w_0^2} I_0 \left(1 - \frac{2x^2(t)}{w_0^2}\right) + o(x^3(t))$: Due to the quadratic nature of the optomechanical coupling at the centre of the beam waist, the piezo-driven displacements create a modulation of the scattered intensity at twice the mechanical resonance frequency, $\delta I_{cal}(t) = -\eta_c \frac{2\sigma_{scatt}}{\pi w_0^2} I_0 \times \frac{x_{cal}^2}{w_0^2} \cos(2\Omega_m t)$. On the other hand, the average detected back-scattered intensity with no drive being applied $\langle I_{bs}^{off}\rangle \simeq \eta_c \frac{2\sigma_{scatt}}{\pi w_0^2} I_0 \times (1 - 2\langle x_{th}^2\rangle/w_0^2)$, with $\langle x_{th}^2\rangle$ the variance of the thermally driven fluctuations of the hybrid CNT resonator. The ratio $\xi_{cal} = \langle\delta I_{cal}^2(t)\rangle^{\frac{1}{2}}/\langle I_{bs}^{off}\rangle$ between the root-mean-square value of the intensity variations at $2\Omega_m$ and the average back-scattered intensity (Fig. 4a) enables to infer

$x_{cal}^2 = \sqrt{2}w_0^2\left(1 - \frac{2\langle x_{th}^2\rangle}{w_0^2}\right)\xi_{cal}.$

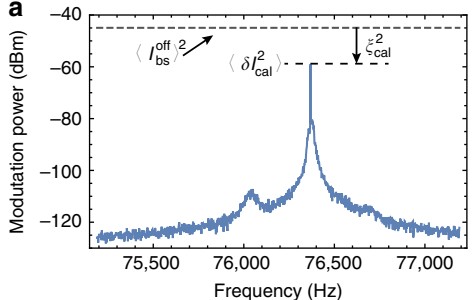
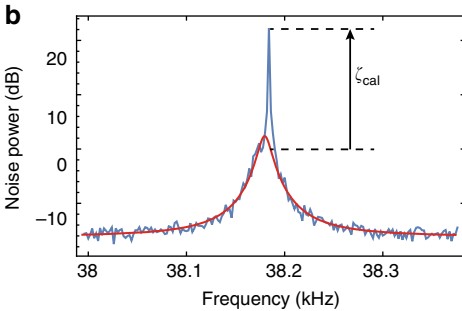

**Fig. 4** Calibration of the nanomechanical motion. **a** Spectrum of the optomechanical response acquired at the centre of the beam waist. The nanomechanical motion is piezo-driven resonantly, resulting in a sharp peak at twice the mechanical resonance frequency. This peak sits on a large noise pedestal resulting from the quadratic optomechanical transduction of the Brownian motion. $\xi_{cal}^2$ ($\simeq 10\,dB$ here) is the ratio between the (DC) power level of the reflected light in absence of any piezo excitation (dashed line) and the driven peak at $2\Omega_m$. **b** Spectrum of the optomechanical response acquired on the side of the beam waist (straight blue line), where linear optomechanical transduction is non zero. The straight, red line corresponds to the Lorentzian fit of associated to thermal noise. $\zeta_{cal}$ ($\simeq 20\,dB$, grossly) corresponds to the ratio between the driven peak (upper dashed line) and the thermal noise level (lower dashed line)

Last, we move the nanomechanical device to the side of the beam waist, where $dI_{bs}/dx \neq 0$. We subsequently evaluate the ratio $\zeta_{cal} = \frac{x_{cal}^2}{2S_{xx}^{th}[\Omega_m]}$ between the power of the intensity modulation resulting from the piezo drive and the thermally driven intensity fluctuations (see Fig. 4b), which identifies to the signal to thermal noise ratio observed on the back-scattered intensity fluctuation spectrum around the mechanical resonance frequency $\Omega_m$. Finally, we obtain

$$S_{xx}[\Omega_m] = \frac{2\langle x_{th}^2\rangle}{\Gamma_m} = \frac{w_0^2\left(1-\frac{2\langle x_{th}^2\rangle}{w_0^2}\right)\xi_{cal}}{\sqrt{2}\zeta_{cal}}, \text{ from which the thermal variance } \langle x_{th}^2\rangle \text{ is}$$

inferred. With $w_0 \simeq 900\,nm$ (see main text), $\xi_{cal} \simeq 0.31$ and $\zeta_{cal} \simeq 135\,Hz$, we find $\langle x_{th}^2\rangle \simeq (293\,nm)^2$.

**Data availability**. The authors declare that the data supporting the findings of this study are available within the article and its Supplementary Information file.

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

## Acknowledgements

The work at ICFO is supported by the ERC advanced grant 692876, the Foundation Cellex, the CERCA Programme, Severo Ochoa (SEV-2015-0522), the grant FIS2015-69831-P of MINECO, and the Fondo Europeo de Desarrollo Regional (FEDER). PV acknowledges support from the French National Research Agency (projects NOFX2015 ANR-15-CE09-0016 and QDOT ANR-16-CE09-0010) and from the ERC starting grant 758794 'Q-ROOT'.

## Author contributions

A.T. has constructed the experimental setup and performed all experiments presented in this work. A.T. and P.V. have proposed the concept of the hybrid CNT resonator. A.S. has conceived the method for selectively growing the Pt nanoparticles on the tips of the CNTs. A.S., and A.T. have developed this method and fabricated the samples used in this work. A.N. has developed the nanopositioning interface used in this work, with advice from A.T. I.T. has assisted with the growth of nanotubes at an early stage of the work. A.T., and P.V. have processed the data presented in this work. P.V. has written the manuscript and Supplementary Information and has performed all calculation. A.B. and P.V. have initiated the project and supervised every step of the work.

## Additional information

**Competing interests:** The authors declare no competing financial interests.

