## [Peer Review File · Nature Communications]

Reviewers' comments:

Reviewer #1 (Remarks to the Author):

The manuscript "Optomechanics with hybrid Carbon Nanotube Resonators" describes optical measurements of the motion of a singly-clamped nanotube mechanical resonator at room temperature. The enabling experimental step is the attachment of a metallic nanoparticle to the end of the nanotube in order to facilitate the optical detection. The authors then characterize the nanotube-particle mechanical properties, quantify its optomechanical coupling, and evaluate how far the fundamental mode is from the quantum regime.

The paper is clearly written, the data appear of high quality, and the analysis is scientifically sound. The submission is also timely in that a number of leading research groups are pushing to study and use bottom-up structures, e.g. nanotubes, graphene sheets, and nanowires, as mechanical resonators and transducers. As the authors point out, these structures -- which have naturally low motional mass -- are also ideal for studying states of motion in the quantum regime. Although the resonator in the paper is far from the quantum regime and no evidence of quantum back-action is shown, it appears to be a promising system, especially given that it can be studied under relatively simple conditions: i.e. at room temperature using a standard laser and detector. For these reasons, I recommend that the paper be published after the authors address the following comments:

1) The metallic nanoparticle is the enabling and therefore crucial component of the author's experiment. In the text I was unable to find details about the size and shape of the particle. In the supplementary section there is an SEM image of a particle on a nanotube, but the resolution is not very good. The authors should at least mention its size in the main text. Also, if possible a higher resolution SEM image of such a particle would be appropriate, given its importance. I realize that such an image may be complicated by the difficulty of making and SEM on a movable nanomechanical object. Also it would be important to mention the mass of the particle, which could roughly be calculated from its observed dimensions. It would be interesting to know its mass in comparison to the nanotube's mass and motional mass.

2) If the mass of the particle is as I suspect, it will be a significant fraction of the motional mass of the nanotube-particle system, if not most of it. In this case, the authors have a mechanical system more akin to a suspended nanoparticle than a carbon nanotube beam. That is, they have a mass secured by a very light string (nanotube). This is also a very interesting system and should be put in the context of works in the direction of suspended nanoparticles (Novotny, et al.). The scheme put forward by the authors may be an excellent way to solve the problem of how to use nanoparticles as

scanning sensors. Such considerations should lead the authors to rethink some of the introductory remarks about similar systems and the relevance of these experiments to other work.

3) At the top of page 3, column 2, the authors state that Γ_p and Ω_p are shown to be linear functions of P in Fig. 3(b). They go on to write that this dependence is correctly described by the equation (2) and the expressions that immediately follow. Nevertheless, in those equations I don't see any clear dependence on P , linear or otherwise. I assume that this can be explained, since F_p is proportional to P , such that $(\partial F_p / \partial x)$ is as well. It would be appropriate, however, for the authors to make these relations more explicit. The previous explanation is crucial for the explanation on "Force sensitivity and quantum limits". Without the knowledge of the exact linear dependence on P , it is not possible to exclude back-action effects or -- more importantly -- to extract the thermal motion and resulting thermally limited force sensitivity.

Reviewer #2 (Remarks to the Author):

The authors demonstrate optical measurement and backaction using a carbon nanotube mechanical resonator. The major claims are novel and interesting, namely:

- Demonstrating a new way to attach an optical mirror to a nanotube.
- Very high force sensitivity, enabled by this advance and by the very low spring constant of a nanotube resonator.
- A new form of cavity-less cooling via the photothermal force.

The work is mostly convincing, but there are some points that should be changed before it is acceptable for Nature Communications.

- 1) A key technical advance is that the authors succeed in growing a nanoparticle by electron beam induced decomposition of a gas precursor. Further technical details are lacking, and essential for reproducibility of the work.
- 2) The authors claim that plausible improvements will "pave the way to quantum coherent manipulation ... all the way up to room temperature." For quantum coherent manipulation, far more than quantum-limited measurement is required; for example strong coupling to a qubit. This claim needs to be removed, or justified a lot more strongly.
- 3) The standard of writing is not careful, and in at least one place makes it impossible to follow the text: the first sentence of the last paragraph on p2 appears meaningless. The opening two

paragraphs of the supplementary, which are essential to understand how the sensitivity is calibrated, are very hard to follow.

4) The paper only describes one resonator, so the title should not be plural.

I would recommend the authors also address the following small errors and points of clarity:

5) It would be helpful to show the axes from Figure 1d also in Figure 2d. It appears that the nanotube points in the $-z$ direction, but this is not clear. It would also help to show the axis of motion in panels c and d, and make clear that it is not aligned exactly with x.

6) On p3, the paper mentions “the scattering force, whose radial component vanishes at the beam waist”. What is this scattering force?

7) There are numerous small errors, including:

a. “carbon nanotubes resonators” (p1)

b. “focal plan” -> “focal plane” (p2)

c. “backaction noises” -> “backaction noise” (p4)

d. “Driven and response and noise” (Fig 2 caption)

e. “correspond are acquired” (Fig. 3 caption)

f. Supplementary line 21: the first minus sign in the equation appears wrong.

g. “piezo-driven resonantly driven” (Fig S1 caption)

h. Fig. S1a: It is not clear what $I_{\text{bs}}^{\text{off}}$ is referring to. How can $\langle I \rangle$ and $\langle I^2 \rangle$ be indicated on the same axes? Since this is a power spectrum, presumably the signal is proportional to I^2 .

Reviewer #3 (Remarks to the Author):

The work by Tavernarakis et al. describes a new approach for making small mass mechanical resonators that can be operated and measured at room temperature, based on a hybrid approach that merges the advantages of nanotube mechanical oscillators and optical probing tools. So far, optics has been instrumental in manipulating mechanical devices down to the quantum limit, but these devices were made by top down lithography and were therefore relatively massive. The approach presented in the current paper offers the possibility to extend the advantages of optical manipulation to the realm of light cantilevers that are based on a carbon nanotube. The authors

demonstrate that using this approach they already obtain significant improvement in the sensitivity of room-T force detection over the state of the art. This could be a significant step, which together with the new experimental setting definitely deserves to be published in Nature Communications, if its importance for room T device operation is justified by the authors (e.g. AFM based on this cantilever, etc.).

It is harder for me to be convinced by the claims that this hybrid device should enable quantum operation at room temperature. Reaching quantum operation means that it should be possible to cool down the resonator to the quantum ground state, to measure it optically with quantum sensitivity and to have it maintain coherence over the time of an experiment. I think that here there are many fundamental issues that the authors ignore. To give two examples:

1. The resonance frequency of the device is few tens of kHz, corresponding to ground state temperature of few micro Kelvin. Reaching that would necessitate active cooling 8 orders of magnitude down from room T. The authors demonstrate some form of cooling, but by less than one order of magnitude, which is far from that. But there is an even more fundamental limitation. Cooling necessarily reduces the quality factor of the resonator. To achieve reasonable Q in the quantum regime the device will have to start at room temperature with quality factors in the billions. The authors have demonstrated a Q factor of few thousands. As far as I know, devices based on nanotubes did not get better than that in room temperature, even the ones that don't have additional mass deposited on them. How do the authors conceive increasing the Q factor by so many orders of magnitude?

2. The authors show that the optomechanical cooling is based on a photo thermal effect, basically a delayed heating caused by the laser. They show that the noise it introduces is comparable to the photonic shot noise and suggest that this is sufficient for cooling down to the quantum ground state. But the problem is that their cooling mechanism is in fact based on heating (at least of part of their resonator). It is possible to see how such mechanism cool a resonator to a few degrees, but it is rather hard to be convinced that a tiny fraction of the heat damped in the platinum blob will not flow to the nanotube, thus precluding the possibility to reach the quantum ground state.

Based on the above I think that the claim of enabling room T quantum operation is weak and that the authors should mostly focus on the new (non quantum) advantages of their device.

There are few more comments that the authors need to address:

3. The measured thermal conductivity of the nanotube turns out to be significantly worse than is reported in the literature. The author suggest that this is because of amorphous carbon shell formed around the nanotube. I cannot see how this shell could make the thermal conductivity worse than in the bare nanotube.

4. Do the authors use single wall nanotube? Multi wall? What is the diameter? What is the spring constant of the resonator? These details are important and are missing from the main text.

5. Can the authors estimate the temperature of the platinum bead during operation? How much power is damped by the laser?

6. Why is the noise in figure 2d only in the x direction and not in the y direction. Shouldn't it be spherically symmetric?

7. There is a typo in the formula of H_{grad}

8. Overall, the paper is very hard to read because it is overloaded with calculations that are more suitable for a supplementary information. It would help the reader if the main text will only refer to them and will give a better description of the big picture.

Point – by – point reply:

We thank all three Referees for their work and careful reading, and overall positive assessment. Below is a point – by – point reply in which we carefully address all the concerns and remarks raised by each Referee. Their questions appear in emphasized character, and our corresponding reply is marked in bold fonts. The corresponding changes are in bold red letters.

Reviewer #1 (Remarks to the Author):

The manuscript "Optomechanics with hybrid Carbon Nanotube Resonators" describes optical measurements of the motion of a singly-clamped nanotube mechanical resonator at room temperature. The enabling experimental step is the attachment of a metallic nanoparticle to the end of the nanotube in order to facilitate the optical detection. The authors then characterize the nanotube-particle mechanical properties, quantify its optomechanical coupling, and evaluate how far the fundamental mode is from the quantum regime.

The paper is clearly written, the data appear of high quality, and the analysis is scientifically sound. The submission is also timely in that a number of leading research groups are pushing to study and use bottom-up structures, e.g. nanotubes, graphene sheets, and nanowires, as mechanical resonators and transducers. As the authors point out, these structures -- which have naturally low motional mass -- are also ideal for studying states of motion in the quantum regime. Although the resonator in the paper is far from the quantum regime and no evidence of quantum back-action is shown, it appears to be a promising system, especially given that it can be studied under relatively simple conditions: i.e. at room temperature using a standard laser and detector. For these reasons, I recommend that the paper be published after the authors address the following comments:

1) The metallic nanoparticle is the enabling and therefore crucial component of the author's experiment. In the text I was unable to find details about the size and shape of the particle. In the supplementary section there is an SEM image of a particle on a nanotube, but the resolution is not very good. The authors should at least mention its size in the main text. Also, if possible a higher resolution SEM image of such a particle would be appropriate, given its importance. I realize that such an image may be complicated by the difficulty of making and SEM on a movable nanomechanical object. Also it would be important to mention the mass of the particle, which could roughly be calculated from its observed dimensions. It would be interesting to know its mass in comparison to the nanotube's mass and motional mass.

We thank the Referee for her/his advice. As pointed by the Referee, imaging our device with the SEM is intrinsically limited by the very large thermal fluctuations, on the order of 300 nm that is several hundred times larger than the typical electron beam size. Therefore, following its "apparent" dimensions and shape, a 575 nm long and 150 nm thick cylinder, one obtains an effective mass of 7.6 picogram, that is 4 orders of magnitude above the calibrated value. Moreover, we have initially chosen not to give geometrical details concerning the platinum nanoparticle as those can only be obtained indirectly for the device reported in our work, as explained below.

To estimate the mass, we rely on the optomechanical analysis reported in our manuscript. From the calibrated displacements, we obtain an effective mass $m_{eff} = 0.79 \times 10^{-18} \text{ kg}$, which is the sum of 3 contributions that are:

- The effective mass of the carbon nanotube resonator itself $m_{CNT} = \pi(r_{CNT}^2 - (r_{CNT} - G)^2)L\rho_C^2/4 \approx 2 \times 10^{-21} \text{ kg}$ (with the $r_{CNT} \approx 0.5 \text{ nm}$ radius of the carbon nanotube resonator, $L \approx 5 \mu\text{m}$ its length $G \approx 0.34 \text{ nm}$ the wall thickness and $\rho_C \approx 2000 \text{ kg m}^{-3}$ the mass density of Carbon).
- The mass of the Platinum nanoparticle m_{Pt} .
- The effective mass of the amorphous carbon shell (see the section ‘Thermal properties and morphology’) $m_{shell} \approx \pi((r_{shell} + r_{CNT})^2 - r_{CNT}^2)L\rho_C/4 \approx 1.5 \times 10^{-19} \text{ kg}$.

The particle mass can subsequently be inferred as $m_{Pt} \approx m_{eff} - (m_{CNT} + m_{shell}) \approx 6.2 \times 10^{-19} \text{ kg}$. As correctly anticipated by the Referee, this represents 80% (resp. 52%) of the motional (resp. physical) mass of the device. Further assuming a cylindrical geometry, the thickness of the nanoparticle can be estimated $t_{Pt} \approx \sqrt{(r_{shell} + r_{CNT})^2 + \frac{m_{Pt}}{\pi L_{Pt} \rho_{Pt}}} - (r_{shell} + r_{CNT}) \approx 1.5 \text{ nm}$, with $L_{Pt} \approx 575 \text{ nm}$ the length of the nanoparticle and $\rho_{Pt} \approx 21450 \text{ kg m}^{-3}$ the mass density of Platinum.

Manuscript changes: We have added a brief statement giving the estimated value of the Platinum nanoparticle’s mass in the manuscript and placed the corresponding details in a new section of the Supplementary Information.

2) If the mass of the particle is as I suspect, it will be a significant fraction of the motional mass of the nanotube-particle system, if not most of it. In this case, the authors have a mechanical system more akin to a suspended nanoparticle than a carbon nanotube beam. That is, they have a mass secured by a very light string (nanotube). This is also a very interesting system and should be put in the context of works in the direction of suspended nanoparticles (Novotny, et al.). The scheme put forward by the authors may be an excellent way to solve the problem of how to use nanoparticles as scanning sensors. Such considerations should lead the authors to rethink some of the introductory remarks about similar systems and the relevance of these experiments to other work.

Manuscript changes: We thank the Referee for her/his remark. We have completed the introduction of our manuscript along the suggestions of the Referee and put our work into context both with respect to nano-optomechanical tweezers and scanning probe experiments, which we support by adequate, new references.

3) At the top of page 3, column 2, the authors state that Γ_p and Ω_p are shown to be linear functions of P in Fig. 3(b). They go on to write that this dependence is correctly described by the equation (2) and the expressions that immediately follow. Nevertheless, in those equations I don't see any clear dependence on P , linear or otherwise. I assume that this can be explained, since F_p is proportional to P , such that $(\partial F_p / \partial x)$ is as well. It would be

appropriate, however, for the authors to make these relations more explicit. The previous explanation is crucial for the explanation on "Force sensitivity and quantum limits". Without the knowledge of the exact linear dependence on P , it is not possible to exclude back-action effects or -- more importantly -- to extract the thermal motion and resulting thermally limited force sensitivity.

We thank the Referee for her/his important comment, to which we fully agree.

Manuscript changes: We have explicit the linear dependence of the photothermal force F_p as a function of the input optical power, yielding to the linear scaling of both dynamical backaction and effective temperature T_{eff} and thereby enables us to conclude to the thermally limited force sensitivity, as correctly emphasized by this Referee.

Reviewer #2 (Remarks to the Author):

The authors demonstrate optical measurement and backaction using a carbon nanotube mechanical resonator. The major claims are novel and interesting, namely:

- Demonstrating a new way to attach an optical mirror to a nanotube.
- Very high force sensitivity, enabled by this advance and by the very low spring constant of a nanotube resonator.
- A new form of cavity-less cooling via the photothermal force.

The work is mostly convincing, but there are some points that should be changed before it is acceptable for Nature Communications.

1) A key technical advance is that the authors succeed in growing a nanoparticle by electron beam induced decomposition of a gas precursor. Further technical details are lacking, and essential for reproducibility of the work.

Manuscript changes: We have added technical details enabling straightforward reproducibility of the work. Note that we are also preparing a manuscript dedicated to the study of the dynamics of the processes involved in the fabrication of our device. These details are not needed for reproducing the reported results and largely exceed the scope of the present work as we believe.

2) The authors claim that plausible improvements will "pave the way to quantum coherent manipulation ... all the way up to room temperature." For quantum coherent manipulation, far more than quantum-limited measurement is required; for example strong coupling to a qubit. This claim needs to be removed, or justified a lot more strongly.

Manuscript changes: We have removed the corresponding statement, along the advice from this Referee.

3) The standard of writing is not careful, and in at least one place makes it impossible to follow the text: the first sentence of the last paragraph on p2 appears meaningless. The opening two paragraphs of the supplementary, which are essential to understand how the sensitivity is calibrated, are very hard to follow.

Manuscript changes: We have amended the manuscript and changed the above referred sentence into an intelligible phrasing.

4) The paper only describes one resonator, so the title should not be plural.

Manuscript changes: We have changed the title along the advice from this Referee.

I would recommend the authors also address the following small errors and points of clarity:

5) It would be helpful to show the axes from Figure 1d also in Figure 2d. It appears that the nanotube points in the $-z$ direction, but this is not clear. It would also help to show the axis of motion in panels c and d, and make clear that it is not aligned exactly with x .

The schematic shown on Fig. 1d was not originally meant to represent our setup in a quantitative manner: Since lasers need to be drawn in the horizontal plane and that our hybrid nanotube resonator is mounted vertically, we initially thought that a more realistic presentation would have been at the expense of clarity. Besides, the Referee is correct to note that the vibrational axis of the hybrid nanotube resonator do not match those of our nano-positioning motor stage. We agree that our previously used notation may be potentially confusing.

Manuscript Changes: We have modified Fig. 1d in order to take into account the comment from this Referee. We have followed the recommendation of this Referee and shown the direction of motion in panel 2c and 2d. We have also adapted the notations in order to avoid any possible confusion.

6) On p3, the paper mentions “the scattering force, whose radial component vanishes at the beam waist”. What is this scattering force?

In general the force exerted by a focused laser beam on a dielectric particle is the sum of two contributions known as the “gradient force” and the “scattering force”, respectively (see Ashkin, et al. "Observation of a single-beam gradient force optical trap for dielectric particles." Optics letters 11.5 (1986): 288-290). The gradient force arises because of the strong variations of the optical power within the beam waist (and therefore of the dielectric force experienced by the particle), whereas the scattering force results from the momentum exchange between light and mechanical motion.

Manuscript changes: We have added a statement specifying the origin of the scattering force in the main manuscript.

7) There are numerous small errors, including:

- a. “carbon nanotubes resonators” (p1)
- b. “focal plan” -> “focal plane” (p2)
- c. “backaction noises” -> “backaction noise” (p4)
- d. “Driven and response and noise” (Fig 2 caption)
- e. “correspond are acquired” (Fig. 3 caption)
- f. Supplementary line 21: the first minus sign in the equation appears wrong.

g. “piezo-driven resonantly driven” (Fig S1 caption)

h. Fig. S1a: It is not clear what I_{bs}^{off} is referring to. How can and be indicated on the same axes? Since this is a power spectrum, presumably the signal is proportional to I^2 .

We thank the Referee for her/his careful reading and recommendations. $\langle I_{bs}^{off} \rangle$ refers to the back scattered intensity in absence of motional drive (i.e. the piezo drive is off). The level indicated on Fig. S1 denotes the corresponding DC power level. The Referee is therefore perfectly correct to expect this level to be proportional to $\langle I_{bs}^{off} \rangle^2$. We have amended the omitted exponent in the new version of the Supplementary Information.

Manuscript changes: We have followed the above stated remarks from this Referee and have amended the manuscript accordingly.

Reviewer #3 (Remarks to the Author):

The work by Tavernarakis et al. describes a new approach for making small mass mechanical resonators that can be operated and measured at room temperature, based on a hybrid approach that merges the advantages of nanotube mechanical oscillators and optical probing tools. So far, optics has been instrumental in manipulating mechanical devices down to the quantum limit, but these devices were made by top down lithography and were therefore relatively massive. The approach presented in the current paper offers the possibility to extend the advantages of optical manipulation to the realm of light cantilevers that are based on a carbon nanotube. The authors demonstrate that using this approach they already obtain significant improvement in the sensitivity of room-T force detection over the state of the art. This could be a significant step, which together with the new experimental setting definitely deserves to be published in Nature Communications, if its importance for room T device operation is justified by the authors (e.g. AFM based on this cantilever, etc.).

It is harder for me to be convinced by the claims that this hybrid device should enable quantum operation at room temperature. Reaching quantum operation means that it should be possible to cool down the resonator to the quantum ground state, to measure it optically with quantum sensitivity and to have it maintain coherence over the time of an experiment. I think that here there are many fundamental issues that the authors ignore. To give two examples:

1. *The resonance frequency of the device is few tens of kHz, corresponding to ground state temperature of few micro Kelvin. Reaching that would necessitate active cooling 8 orders of magnitude down from room T. The authors demonstrate some form of cooling, but by less than one order of magnitude, which is far from that. But there is an even more fundamental limitation. Cooling necessarily reduces the quality factor of the resonator. To achieve reasonable Q in the quantum regime the device will have to start at room temperature with quality factors in the billions. The authors have demonstrated a Q factor of few thousands. As far as I know, devices based on nanotubes did not get better than that in room temperature, even the ones that don't have additional mass deposited on them. How do the authors conceive increasing the Q factor by so many orders of magnitude?*

2. The authors show that the optomechanical cooling is based on a photo thermal effect, basically a delayed heating caused by the laser. They show that the noise it introduces is comparable to the photonic shot noise and suggest that this is sufficient for cooling down to the quantum ground state. But the problem is that their cooling mechanism is in fact based on heating (at least of part of their resonator). It is possible to see how such mechanism cool a resonator to a few degrees, but it is rather hard to be convinced that a tiny fraction of the heat damped in the platinum blob will not flow to the nanotube, thus precluding the possibility to reach the quantum ground state.

Based on the above I think that the claim of enabling room T quantum operation is weak and that the authors should mostly focus on the new (non quantum) advantages of their device.

We agree with the comment of this Referee that a number of conditions need to be further matched with our system before granting quantum *mechanical* operation at room temperature. In particular, this Referee correctly points that the combination of very high phonon occupation, insufficient mechanical Q – factor, and optical absorption make unlikely to coherently cool down to the zero – point fluctuation level at room temperature with the device reported in our manuscript.

However and importantly, we reemphasize that there is an additional condition for enabling quantum operation: This condition is the operation close to the Heisenberg limit that is with a measurement uncertainty product much larger than $\hbar/2$ (as pointed by this Referee in her/his comment). This crucial aspect is one of the main reasons why attaining the quantum groundstate remains so difficult by pure laser (or microwave) cooling means. This aspect is even much reinforced for bottom up devices, which have so far been shown to suffer from critical decoherence mechanism, preventing to reach ultra-low phonon occupancies. In our work, we characterize the quantum measurement efficiency and find that simple technical improvements should enable to operate close to the Heisenberg limit. We initially did not expect such result and believe it is remarkable, opening the realistic perspective to use bottom up ultra-light devices for ultra-sensitive optomechanical applications.

To address more specifically the questions and comments of the Referee:

1) The Referee raises two points that make questionable the ability to manipulate our hybrid nanomechanical system at the quantum level. A) This Referee notes that the mechanical resonance frequency of our reported system remains quite low, in the few tens of kHz. This corresponds to extremely low groundstate temperatures, in the few micro Kelvin range, which are very difficult to attain even using active cooling means. Importantly, our approach does not limit to such low frequency range, which was selected due to the exquisite room temperature force sensitivity in the frame of the present work. We have been able to measure and characterize resonators up to the 10 MHz range and with similar mechanical Q factors, which are very promising and will relax cooling conditions from around 3 orders of magnitude. B) Related to the Q factor, this Referee points that it will enable only limited cooling rate (on the order of the Q factor), insufficient to reveal quantum behaviours. Indeed, as this Referee seems to correctly guess, additional measures must be taken to fully exploit the reported exquisite force sensitivity of our system. In particular, coupling the hybrid nanomechanical device to a high finesse optical cavity, thereby forming a cavity nano-optomechanical system, seems an extremely promising way. We have written some of

the perspectives of our work along these lines, implicitly referring to recent work^{1,2}. However, we admit that more details would be needed for making these claims more convincing.

2) In the second point of her/his comment, the Referee criticizes the fact that our reported cooling results are based on a delayed *absorption* process, which comes along heating of the device. Once more this Referee is correct, optical absorption (closely related to decoherence) is a ubiquitous source of limitation for coherent laser cooling. In fact, the quantum perspectives raised in our conclusion do not refer to using our system as such for reaching the quantum groundstate, but rather implementing it into a cavity optomechanical design, once the (swift) suggested improvement completed (see the last section of our manuscript), notably using a much less absorbing material. Such configuration will grant both much increased sensitivity and cooling rate via a delayed gradient force dynamical backaction mechanism.

Manuscript changes: In light of the above, we have amended our conclusion and removed the statement regarding the perspective of quantum coherent manipulation of our device at room temperature. Additionally, we have modified both introduction and conclusion for better highlighting the potential of our system for room temperature ultra-sensitive applications such as scanning probe microscopy, magnetic imaging, force sensing etc. which we substantiate using appropriate references.

There are few more comments that the authors need to address:

3. *The measured thermal conductivity of the nanotube turns out to be significantly worse than is reported in the literature. The author suggest that this is because of amorphous carbon shell formed around the nanotube. I cannot see how this shell could make the thermal conductivity worse than in the bare nanotube.*

In our work, we interpret the decreased thermal conductivity of the nanotube resonator as a possible consequence of the formation of an amorphous carbon shell around the nanotube, acting as an additional thermal resistance, placed in parallel of that of the bare nanotube (i.e. some of the incoming thermal flux is distributed to the amorphous carbon shell). As described in the supplementary information of our manuscript, this hypothesis reasonably agrees with independent imaging and dynamical measurements reported in a previous work, consistent with an amorphous carbon layer thickness of a few nanometres. Note that alternative explanations may also be envisioned, disorder or contamination along the nanotube which might be produced during the electron beam-induced growth of the nanoparticle.

Manuscript changes: We have added a statement that alternative phenomena may also contribute to the observed decreased thermal conductance.

4. *Do the authors use single wall nanotube? Multi wall? What is the diameter? What is the spring constant of the resonator? These details are important and are missing from the main text.*

In our work, we infer the effective physical properties of our hybrid carbon nanotube resonator from the measured nanomechanical motion which enable to extract both effective (motional mass) $m_{eff} \approx 0.79 \times 10^{-18} kg$ and mechanical resonance frequency $\frac{\Omega_m}{2\pi} \approx 38178.5 Hz$ yielding an effective spring constant $k_{eff} = m_{eff}\Omega_m^2 \approx 4.5 \times 10^{-8} Nm^{-1}$. These properties reflect those of the whole device, including the platinum nanoparticle and possible residual amorphous carbon deposition (as commented above). Selective extraction of the physical properties of the nanotube itself requires running specific characterization (electron microscopy, Raman spectroscopy...) which we did not proceed with the sample reported in our manuscript. For this reason, we preferred not to give physical details related to the bare carbon nanotube resonators. Our experience is that our growing process essentially yields to carbon nanotubes with typical diameter in the 1 – 3 nm range. Some of the nanotubes can be double and triple-walled. Importantly, our intensive work on nanotube resonators over the last years indicates that the diameter and the number of walls have little influence on the mechanical properties compared to the mechanical tension and the adsorbed contamination.

Manuscript changes: We have added a statement that our growing method essentially yields nanotubes with diameters in the above stated range.

5. Can the authors estimate the temperature of the platinum bead during operation? How much power is damped by the laser?

We thank the Referee for her/his interesting question. The temperature of the platinum nanoparticle crucially depends on the *nature of the thermal contact* between the particle and the carbon nanotube. The current state of our knowledge only leaves us with two possible extreme hypotheses that are either perfect thermal contact or absence of any thermal contact. Obviously, these two situations correspond to radically different steady state temperatures, from very moderate values in presence of perfect thermal contact (where the incoming thermal flux is fully transmitted to the wafer which acts as a heat sink), to very high values (in excess of several hundred K) in absence of any contact (because of the low density of the thermalizing medium, that is the residual air present in the vacuum chamber). In both cases, the resulting effect on the overall temperature of the device is expected to be negligible, which correspond to what we observe in our experiment as explained in our manuscript: Thus, our reported results do not enable to settle the question of the temperature of the platinum nanoparticle, which would require additional experimental work that is beyond the scope of the present study.

Regarding the second part of the question of this Referee, the effective thermomechanical power damped by the laser can be estimated as $P_F^{cool} \approx \Gamma_p k_m \langle x^2 \rangle = \frac{\Gamma_m \Gamma_p}{\Gamma_m + \Gamma_p} \times \frac{k_B T}{m_{eff} \Omega_m^2} \approx 4 \times 10^{-19} W$ (with Γ_m the intrinsic damping rate, Γ_p the optically induced damping rate, k_B Boltzmann's constant, T the temperature of the thermal bath, m_{eff} the motional mass, Ω_m the mechanical resonance frequency and $k_m = m_{eff} \Omega_m^2$ the spring constant) for an incident optical power of 20 μW .

6. Why is the noise in figure 2d only in the x direction and not in the y direction. Shouldn't it be spherically symmetric?

Figure 2d is obtained by plotting the noise spectral density measured at the mechanical resonance frequency associated with the mechanical motion in direction x. This mechanical mode is dithering the position of the hybrid nanomechanical resonator tip in the same direction, which creates a modulation $\delta I[\Omega_m] \simeq \left(\frac{\partial I}{\partial x}\right)_{(x_0, y_0, z_0)} \delta x[\Omega_m]$, with (x_0, y_0, z_0) the averaged position within the beam waist. Since the thermal fluctuations are independent of (x_0, y_0, z_0) , the intensity noise directly reflects the sensitivity factor $\left(\frac{\partial I}{\partial x}\right)_{(x_0, y_0, z_0)}$, which corresponds to what is observed on Fig. 2d.

Besides, at the beam waist, the motion in direction y is not coupled to light $\left(\frac{\partial I}{\partial z}\right)_{(x_0, y_0, z_0)} = 0$, and the optomechanical fluctuations measured around the mechanical resonance frequency associated with mechanical motion in direction z are expected to be zero, which is observed in our experiment.

7. There is a typo in the formula of H_{grad}

We did not identify the mistake to which the Referee is referring to. In our manuscript, we write that $H_{grad}[\Omega] \simeq 1 \forall \Omega$, which means that H_{grad} is constant, equal to unity for at all mechanical frequency.

Manuscript changes: We have removed the possibly litigious symbol for clarity.

8. Overall, the paper is very hard to read because it is overloaded with calculations that are more suitable for a supplementary information. It would help the reader if the main text will only refer to them and will give a better description of the big picture.

As pointed by this Referee, we have chosen to include many details in our manuscript to avoid too much referring to the supplementary information, which can be detrimental for an accurate understanding of our work from a non-specialized readership. We believe that under such form, the reader has essential information enabling self-consistent reading of the manuscript. Additionally, we believe that such amount of details is required for clearly stating our hypothesis and models, which is important in view of both validating our approach and reproducing the work. Consequently, we prefer to maintain most of our presentation as such.

References

- 1 Norte, R. A., Moura, J. P. & Gröblacher, S. Mechanical resonators for quantum optomechanics experiments at room temperature. *Physical review letters* **116**, 147202 (2016).

- 2 Reinhardt, C., Müller, T., Bourassa, A. & Sankey, J. C. Ultralow-Noise SiN Trampoline Resonators for Sensing and Optomechanics. *Physical Review X* **6**, 021001 (2016).

Reviewers' comments:

Reviewer #1 (Remarks to the Author):

The authors have addressed each referee's comments in turn. I find their responses satisfactory and suggest publication of this manuscript.

Reviewer #2 (Remarks to the Author):

The authors have satisfactorily addressed all my concerns and I recommend acceptance.

I do request that the authors pay attention to the vertical axis in Fig. S1 (b). I think the units should be dB instead of Log.

Reviewer #3 (Remarks to the Author):

The authors have addressed most of the issues raised. They gave a detailed answer in the response letter, which put things in better perspective. However, they have hardly included any of this discussion in the actual manuscript. I think that it is important to include the main points that are well explained in the reply letter also in the paper. Specifically, it would be important to include in the main text the following points that are discussed in the reply letter:

1. A discussion regarding the difference between reaching the quantum ground state and the Heisenberg limit of detection, and what are the different consequences in terms of possible future experiments.

2. Mention that with the current parameters, the existing devices cannot reach the quantum limit, but if put in a cavity this might be feasible (especially since in the abstract and summary there are still statements about using this device for quantum mechanical operations at room temperature).

3. Describe more quantitatively the effect of heat absorption by the platinum bid and how this will limit the cooling of the mechanical mode.

Also, I'm still confused by the answer to my question about the thermal conductivity through the nanotube – if you add a resistor in parallel to another resistor, how can it increase the total resistance? (which, if I understand correctly, is what the authors claim is happening to the thermal resistance due to the addition of a parallel shell of amorphous carbon).

Point – by – point reply:

Once more, we thank all three Referees for their work and help in improving the presentation of our manuscript. In particular, we thank Referees #1 and Referee #2 for their explicit recommendation to publish our work in *Nature Communications*. We also thank Referee #3 for his very positive assessment. Below is a point – by – point reply in which we address the remaining minor points Referee #3, which appear in emphasized character. Our corresponding reply is marked in bold fonts. The corresponding changes are in bold red letters.

Reviewer #3 (Remarks to the Author):

The authors have addressed most of the issues raised. They gave a detailed answer in the response letter, which put things in better perspective. However, they have hardly included any of this discussion in the actual manuscript. I think that it is important to include the main points that are well explained in the reply letter also in the paper. Specifically, it would be important to include in the main text the following points that are discussed in the reply letter:

1) *A discussion regarding the difference between reaching the quantum ground state and the Heisenberg limit of detection, and what are the different consequences in terms of possible future experiments.*

2) *Mention that with the current parameters, the existing devices cannot reach the quantum limit, but if put in a cavity this might be feasible (especially since in the abstract and summary there are still statements about using this device for quantum mechanical operations at room temperature).*

Manuscript changes: We have added a discussion paragraph including the distinction between reaching the quantum groundstate by coherent cooling means and Heisenberg limited optomechanical measurement, as well as a statement regarding future required improvement for room temperature quantum mechanical operation.

3) *Describe more quantitatively the effect of heat absorption by the platinum bid and how this will limits the cooling of the mechanical mode.*

Manuscript changes: We have added a paragraph at the end of the subsection “Thermal properties and morphology” which emphasizes the dissipative nature of the photothermal interaction and the corresponding limits for coherent cooling.

4) *Also, I’m still confused by the answer to my question about the thermal conductivity through the nanotube – if you add a resistor in parallel to another resistor, how can it increase the total resistance? (which, if I understand correctly, is what the authors claim is happening to the thermal resistance due to the addition of a parallel shell of amorphous carbon).*

We agree with the Referee that adding two resistors in parallel cannot result into a net increase of the total resistance. However this does not hold true for the resistivity, which is proportional to the cross section area of the sample. Hence, as demonstrated in our Supplementary Section 2 “Thermal Properties and Morphology”, the inferred amorphous carbon shell cross section area, about 100 times that of the pristine tube, justifies the observed strong increase (resp. decrease) of the hybrid nanotube resistivity (resp. conductivity).